

# The Ozone Mapping and Profiler Suite (OMPS) Limb Profiler (LP) Version 1 Aerosol Extinction Retrieval Algorithm: Theoretical Basis

Robert Loughman[1], Pawan K. Bhartia[2], Zhong Chen[3], Philippe Xu[4], Ernest Nyaku[1], and Ghassan Taha[5]

[1]Department of Atmospheric and Planetary Sciences, Hampton University, Hampton, Virginia, USA
[2]Atmospheric Chemistry and Dynamics Laboratory, NASA Goddard Space Flight Center, Greenbelt, Maryland, USA
[3]Science Systems and Applications, Inc. (SSAI), 10210 Greenbelt Road, Suite 600, Lanham, Maryland 20706, USA
[4]Science Applications International Corporation (SAIC), Lanham, Maryland, USA
[5]GESTAR, Columbia, Maryland, USA

*Correspondence to:* Robert Loughman (robert.loughman@hamptonu.edu)

**Abstract.**

The theoretical basis of the Ozone Mapping and Profiler Suite (OMPS) Limb Profiler (LP) Version 1 aerosol extinction ($AE$) retrieval algorithm is presented. The algorithm uses an assumed bi-modal log-normal aerosol size distribution to retrieve $AE$ profiles at 675 nm from OMPS LP radiance measurements. A first-guess $AE$ profile is updated by iteration using the

Chahine non-linear relaxation method, based on comparisons between the measured radiance profile at 675 nm and the radiance profile calculated by the Gauss-Seidel Limb Scattering (GSLS) radiative transfer model for a spherical-shell atmosphere. This algorithm is discussed in the context of previous limb-scattering $AE$ retrieval algorithms, and the most significant error sources are enumerated. The retrieval algorithm is limited primarily by uncertainty about the aerosol phase function, and by horizontal variations in aerosol extinction, which violate the spherical-shell atmosphere assumed in the Version 1 algorithm.

## 1    Introduction

Most of the aerosols found in the Earth's atmosphere occur in the planetary boundary layer, due to the wide variety of aerosol sources that exist at the surface (dust, smoke, sea salt, etc.). But a secondary peak in aerosol abundance typically occurs in the stratosphere (Junge et al., 1961a), extending from the tropopause to an altitude of approximately 30 km (Brock et al., 1995 ; Hamill et al., 1997). The stratospheric aerosol layer consists primarily of sulfuric acid ($H_2SO_4$) droplets, generated

by the oxidation of tropospheric sulfur dioxide ($SO_2$) and carbonyl sulfide ($OCS$) that has entered the stratosphere through troposphere-stratosphere exchange processes (Holton et al., 1995). The stratospheric aerosol layer is enhanced by volcanic eruptions that inject $SO_2$ into the stratosphere, creating a layer of $H_2SO_4$ droplets that spreads horizontally and slowly dissipates over a period from months to several years. Volcanic eruptions also may inject ash particles directly into the stratosphere, and mineral dust from the ablation of meteors also can augment the stratospheric aerosol layer (Cziczo et al., 2001). Sev-

eral competing influences therefore affect the stratospheric aerosol layer, including volcanic activity, stratosphere-troposphere exchange, stratospheric transport processes, gas-to-droplet conversion rates, and particle sedimentation. As a result, the stratospheric aerosol concentration varies widely in space and in time, as shown in Fig. 1.





Aerosols in the stratosphere play key roles in the chemistry of that region, particularly including heterogeneous ozone destruction (Hofmann and Solomon, 1989; McCormick et al., 1995; Meinrat and Crutzen, 1997; Solomon, 1999). Monitoring stratospheric aerosols as a tracer for stratospheric air mass motion has also provided useful insight (Holton et al., 1995; Goering et al., 2001). The most significant climate impact of changes in the distribution of stratospheric aerosols occurs due to back-

scattering of solar radiation, which increases the planetary albedo and cools the troposphere (Robock, 2000; Kravitz et al., 2011; Ridley et al., 2014). The magnitude of this effect varies significantly with latitude, time of day, etc. (Deshler et al., 2008). A recent review of the observations and processes of stratospheric aerosol and how they impact the Earth's climate is presented in (Kremser et al., 2016).

## 1.1  Occultation measurements

The primary global record of stratospheric aerosol abundance has been derived from solar occultation (SO) data. The SAM / SAGE series of missions pioneered this technique, with the long-lived SAGE II instrument (1984-2005) providing a particularly valuable continuous data record (Russell and McCormick, 1989; McCormick and Veiga, 1992; Thomason et al., 1997). These SO measurements provide unmatched altitude resolution, precision and accuracy for stratospheric aerosol monitoring: Transmission profiles are produced on a 0.5 km grid with estimated vertical resolution = 0.7 km (SAGE, 2002), while providing

$5\%$ targeted accuracy and precision for aerosol extinction coefficient $AE$ (Thomason et al., 2010). The POAM satellite (Lucke et al., 1999) series has further provided SO measurements in the polar regions. Comparison between POAM III and SAGE II data indicates relative differences of $\pm 30\%$ in $AE$, with some hemispheric differences evident (Randall et al., 2001). The MAESTRO instrument also launched aboard the SCISAT satellite in 2003 (McElroy et al., 2007). This mission has provided aerosol extinction profiles based on SO measurements, as described by Sioris et al. (2010) and McElroy (2016).

The primary drawbacks of SO observations made from a low-Earth orbit are the limited number of profiles measurable (24 occultations per day), and the lack of flexibility concerning the locations monitored (which are determined entirely by the orbit of the satellite). In addition to SO measurements, occultation measurements involving other sources of light are also possible. The SAGE III instrument also performs lunar occultations, but does not produce $AE$ profiles based on lunar occultation measurements (Thomason et al., 2010). The GOMOS instrument (Bertaux et al., 2010) has provided stellar occultation monitoring

of the stratospheric aerosol layer (Vanhellemont et al., 2016). Since numerous bright stars can be used as the source of photons, this method offers the potential for increased geographic coverage than SO (but with a much dimmer source of light). Comparisons of GOMOS stellar occultation $AE$ retrievals to SAGE II, SAGE III and POAM III $AE$ data indicate agreement at the $10 - 25\%$ level in the lower stratosphere (Vanhellemont et al., 2010).

The lack of global stratospheric $AE$ profile measurements from SO since the SAGE II, POAM III and Meteor-3M SAGE

III missions ended (in 2005, 2005 and 2006, respectively) has left a vacancy. Limb scattering (LS) data has been combined with occultation data (Rieger et al., 2015) to produce a merged time series, which will aid in tracking the evolution of aerosol plumes from volcanic eruptions that contribute aerosol to the upper troposphere and lower stratosphere (UTLS) (Andersson et al., 2015). After an absence of over a decade, the recent installation of a SAGE III instrument on the International Space Station (Cisewski et al., 2014) in February 2017 promises to resume the valuable SO dataset for stratospheric $AE$ monitoring.



## 1.2 Limb Scattering (LS) measurements

Several recent missions have provided LS measurements, including the Optical Spectograph and InfraRed Imaging System (OSIRIS) (Llewellyn et al., 2004), the Scanning Imaging Absorption spectroMeter for Atmospheric CartograpHY (SCIA-MACHY) (Bovensmann et al., 1999), Meteor-3M SAGE III (Mauldin et al., 1998) (which made LS measurements in addition to occultation measurements), and the Ozone Mapping and Profiler Suite, Limb Profiler (OMPS LP) (Flynn et al., 2006). These instruments measure profiles of the LS sunlight across the ultraviolet (UV), visible and near infrared (NIR) spectral regions.

As illustrated in Fig. 2, LS measurements are possible throughout the entire sunlit hemisphere, permitting much better spatial coverage and sampling than SO measurements. But LS retrievals of stratospheric $AE$ are significantly more challenging, requiring careful tangent height registration of the measured radiance profiles (Moy et al., 2017) and cloud screening (Chen et al., 2016). The LS radiance is also susceptible to stray light contamination (see Fig. 2 of Rault (2005)). Finally, the LS radiance depends upon both the scattering properties (especially the phase function) and the extinction coefficient for the aerosols, while occultation measurements are only sensitive to the latter property.

Each LS mission team has developed its own methodology to retrieve stratospheric $AE$ profiles from limb radiance measurements, but all of the retrieval algorithms involve the comparison of measured LS radiance profiles with simulated radiance profiles that are generated by a radiative transfer (RT) model. In the case of OSIRIS, the "color index" of measured LS radiances at 470 and 750 nm are compared to radiances calculated by the SASKTRAN (Bourassa et al., 2008a; Zawada et al., 2015) model. The evolution of $AE$ during the OSIRIS mission has been investigated in a series of papers (Bourassa et al., 2007; Bourassa et al., 2010; Bourassa et al., 2012). Comparison between Version 5 OSIRIS retrievals and the Version 4 SAGE III record indicates agreement to within $10\%$ for $AE$ in the 15-25 km altitude range (Bourassa et al., 2012). The retrieval of aerosol size information from OSIRIS data has also been investigated (Bourassa et al., 2008b; Rieger et al., 2014) to produce the Version 6 OSIRIS aerosol product. The Version 6 algorithm combines the Infrared Imager 1.53 $\mu m$ channel with OSIRIS data to allow retrieval of both $AE$ and aerosol mode radius, based on an assumed aerosol mode width value.

For the SCIAMACHY mission, the initial $AE$ retrievals were performed by Taha et al. (2011), using a modified version of the algorithm under development for the eventual OMPS LP mission (Rault and Loughman, 2013). Ovigneur et al. (2011) present an approach to retrieve stratospheric aerosol number density from SCIAMACHY LS data in the $O_2$ A-band. More recent work (Ernst et al., 2012; Ernst, 2013; Von Savigny et al., 2015) describes an approach that uses the color-index approach introduced by (Bourassa et al., 2007). The global average difference between SAGE II (Version 7) and SCIAMACHY (Version 1.1) $AE$ data is $10\%$, with larger relative differences (up to $40\%$) at specific latitudes and altitudes (Von Savigny et al., 2015). The SCIATRAN RT model (Rozanov et al., 2014) provides the radiance simulations in this case.

The SAGE III instrument that flew on the Meteor-3M satellite made LS measurements as a research product, from which retrievals of ozone (Rault, 2005) and aerosol (Rault and Loughman, 2007) were derived. These retrieval algorithms were the predecessors for the initial OMPS LP algorithm (Rault and Loughman, 2013), which used the GSLS RT model described in Loughman et al. (2004) to provide the simulated radiances. Comparison to coincident SAGE II SO data indicated bias $< 5\%$ and precision $= 25 - 50\%$ for $AE$ retrievals from SAGE III LS data (Rault and Loughman, 2007).





The $AE$ retrieval algorithm described by Rault and Loughman (2013) was applied to early OMPS LP observations. It was modified slightly to assess the aftermath of the Chelyabinsk bolide explosion, as documented by Gorkavyi et al. (2013). This paper describes the new OMPS LP Version 1 (V1) $AE$ retrieval algorithm. Section 2 briefly describes the OMPS instruments (particularly the LP instrument) and the Suomi NPP (SNPP) satellite on which OMPS was initially installed. Section 3 focuses

on the necessary radiance calculations, while Section 4 describes the retrieval algorithm in detail. Section 5 contains error analysis of the retrieved aerosol extinction profiles. Finally, a preliminary evaluation of the retrieval results is presented in Section 6. We conclude with a summary and description of proposed future work in Section 7.

## 2    The OMPS LP Instrument

The LP instrument is part of the Ozone Mapping and Profiler Suite (OMPS), whose primary purpose is to monitor the ozone
layer. The LP instrument design was guided by the preceding SOLSE and LORE sensors (McPeters et al., 2000) and was built by Ball Aerospace Technology Corporation under contract from the Integrated Program Office. The instrument makes a series of simultaneous observations of the Earth's entire sunlit limb through three vertical slits, producing a set of three radiance profiles: The line of sight (LOS) for one set of observations (called the "center slit") is oriented along the orbital track, while the other two sets (called the "left" and "right" slits) are offset by $4.25°$ from the orbital track. The ground track of the resulting
sequence of observations is illustrated in Fig. 3.

OMPS LP is installed in a fixed orientation relative to the SNPP spacecraft, which is in a sun-synchronous orbit with a 1:30 PM ascending node and mean altitude = 833 km above the Earth's surface. As a result of this orientation, the single scattering angle ($SSA$) observed by the LP instrument varies with latitude as shown in Fig. 4. Most notably, Northern Hemisphere observations (with latitude $> 0°$) generally correspond to forward-scattered beams ($SSA < 90°$), while Southern Hemisphere
observations (latitude $< 0°$) correspond to back-scattered beams ($SSA > 90°$). As a result, the relative strength of the aerosol scattering signal is much larger in Northern Hemisphere OMPS LP measurements, as shown in Fig. 5: The aerosol phase function ($APF$) increases by a factor of approximately 50 over the course of a typical orbit, as the SNPP satellite travels from its southernmost observation to its northernmost observation. (All observations for which the solar zenith angle at the tangent point $\theta_T < 85°$ are processed by the OMPS LP V1 software.)

The OMPS LP instrument permits radiance observations for the $290 - 1000$ nm wavelength range. Dispersion is provided by a prism, which provides images whose spectral resolution varies greatly with wavelength (from $\approx 1$ nm in the UV to $\approx 30$ nm in the NIR). For further information about the OMPS LP instrument characteristics, please consult Flynn et al. (2006), Rault and Loughman (2013) and Jaross et al. (2014).



## 3 Radiance Calculation

### 3.1 The GSLS Radiative Transfer Model

The GSLS RT model is built from the previous models described by Herman et al. (1994) and Herman et al. (1995)), as summarized in Loughman et al. (2004). The model atmosphere is specified by input pressure, temperature, absorbing gas number density, and $AE$ profiles. Cross-sections for Rayleigh scattering and gaseous absorption are provided by the user for the wavelengths of interest. The aerosol scattering and absorption properties are calculated using Mie theory, given the user-provided aerosol microphysical and optical properties. The viewing geometry is specified by the solar zenith angle and relative azimuth angle at the tangent point (TP) for the LOS, denoted by $\theta_T$ and $\phi_T$, respectively, and illustrated in Fig. 6.

The GSLS model calculates radiances at several wavelengths $\lambda$ and tangent heights $h$. For single-scattering (SS) calculations, the solar beam attenuation is calculated to each point along the LOS, including the curvature of the spherical atmosphere as well as the variation of solar zenith angle and solar beam attenuation along the LOS. The attenuation of the scattered beam along the LOS is also calculated accounting for the curvature of the atmosphere. Recent updates to the GSLS model described in Loughman et al. (2015) reduce SS radiance errors that were as great as $4\%$ in the Loughman et al. (2004) comparisons to the $0.3\%$ level.

The multiple scattered (MS) radiances observed by a LS instrument originate from illumination of the limb LOS by photons that have been scattered within the atmosphere or reflected by the underlying surface. These photons are scattered for the final time at some point along the limb LOS, and then transmitted from that point to the observer. The diffuse upwelling radiance ($DUR$) from below the LOS provides the primary source of illumination that produces MS photons, containing the combined effects of molecular scattering, aerosol scattering, cloud scattering, and surface reflection. For the V1 $AE$ retrieval, the $DUR$ is estimated as described in Sect. 3.2.

The MS source function is calculated at one or more points along the LOS using the pseudo-spherical version of the RT model described by Herman et al. (1994) and Herman et al. (1995). In the Loughman et al. (2004) GSLS model, the MS source functions were calculated only at the TP (solar zenith angle = $\theta_T$). This was updated in Loughman et al. (2015) to calculate the MS source functions at multiple solar zenith angles along the LOS, increasing the accuracy of the MS radiances. Total radiance errors that had reached $10\%$ in the Loughman et al. (2004) comparisons decline to $1-3\%$ in the updated comparisons presented by Loughman et al. (2015).

The GSLS model described by Loughman et al. (2004) was used for retrieval applications on missions including the Shuttle Ozone Limb Sounding Experiment (SOLSE) / Limb Ozone Retrieval Experiment (LORE) (Flittner et al., 2000), SAGE III (Rault, 2005; Rault and Taha, 2007; Rault and Loughman, 2007), GOMOS (Taha et al., 2008), SCIAMACHY (Taha et al., 2011) and OMPS LP (Rault and Loughman, 2013). These retrieval algorithms generally performed well despite the shortcomings of the Loughman et al. (2004) version of the GSLS model, but development of a more accurate version of the GSLS model was considered desirable to improve the algorithms further, as well as for the purpose of interpreting residuals (differences between measured radiances and radiances calculated for the desired model atmosphere). The Loughman et al. (2015) version of GSLS has therefore been implemented for the V1 algorithm described in this paper.



## 3.2 The Diffuse Upwelling Radiance ($DUR$)

The horizontal extent of the limb LOS covers thousands of kilometers, and the underlying scene generally includes variable surface types, broken clouds at various locations and levels, etc. The current GSLS model lacks the capability to model the full complexity of such a scene, even if its properties were known. To estimate the $DUR$, the V1 $AE$ retrieval algorithm uses

a simple Lambertian model of the reflecting surface, characterized by its reflectivity $R$. Radiances simulated by the GSLS RT model using a Lambertian surface (placed at sea level) are used to estimate an effective scene reflectivity from a measurement, by tuning the value of $R$ used in the GSLS model until the calculated radiance matches the measured value for a given set of viewing and illumination conditions.

    The $R$ value at which the calculations match the measurement is sometimes called the "Lambert-equivalent reflectivity"

or $LER$. It does not equal the true reflectivity of the surface, since the scene generally contains clouds, aerosols, etc. below the LOS that are not properly captured in the GSLS model atmosphere, and variations in terrain height are also ignored. This approach has been extensively used for nadir-viewing applications such as ozone profile retrievals from the SBUV satellite series and ozone total column retrievals from the TOMS satellite series (Heath et al., 1975), and was suggested by Mateer et al. (1971). Approximate treatment of $DUR$ in the V1 OMPS LP $AE$ retrieval algorithm is justified by the relative insensitivity

of the normalized radiances used by the $AE$ retrieval to $DUR$, as demonstrated in Fig. 11.

    Finally, note that the model atmosphere for the GSLS model used in the V1 $AE$ retrieval algorithm is constrained to be 1-dimensional (i.e., the atmospheric properties vary only with altitude). A 2-dimensional SS version of GSLS (allowing atmospheric properties to vary along the LOS as well as with altitude) has recently been developed (Loughman et al., 2016), and a full MS version of this model is currently under development.

## 20  3.3  Aerosol Properties

The LS radiance is affected by several aerosol properties. The V1 algorithm described in this paper employs assumptions for several of these properties in order to deduce the $AE$ based on observations of the LS radiance $I(\lambda, h)$.

### 3.3.1  Aerosol Shape and Optical Properties

First, the stratospheric aerosols are assumed to be spherical droplets of sulfuric acid ($H_2SO_4$). Mie theory is used to calculate

the aerosol scattering and extinction properties, based on the aerosol refractive index values given in Table 1. These assumptions exclude numerous processes that may contribute significantly to the stratospheric aerosols found at particular places and times (e.g., volcanic ash, meteoric dust, various tropospheric aerosols that enter the stratosphere). However, the assumption that "aged" aerosol in the Junge layer is dominated by such $H_2SO_4$ droplets agrees with observations dating back to the earliest studies of stratospheric aerosol (Junge et al., 1961a), and is assumed in all previous LS $AE$ retrieval algorithms. The assump-

tion is less supportable under "perturbed" stratospheric conditions (such as the immediate aftermaths of volcanic eruptions), as noted by Vernier et al. (2016), or at the upper and lower boundaries of the Junge layer, which may have more meteoric content above and more tropospheric aerosol near the tropopause.





### 3.3.2 Aerosol Size Distribution (ASD)

In the V1 algorithm, the ASD is modeled as a bi-modal log-normal (LN) distribution, as specified in Table 1. This ASD is defined by equation ( 1):

$$\frac{dN(r)}{dr} = \sum_{i=1}^{2} \frac{N_i}{r\sqrt{2\pi}\log\sigma_i} \exp\left\{-\frac{1}{2}\left[\frac{\log(r/r_i)}{\log\sigma_i}\right]^2\right\} \tag{1}$$

Five independent parameters are required to specify the shape of the bi-modal LN ASD: 2 mode radii ($r_1$ and $r_2$), 2 mode widths ($\sigma_1$ and $\sigma_2$) and 1 more parameter indicating the relative sizes of the aerosol concentration associated with each mode ($N_1$, $N_2$). In this work, the mode with the smaller mode radius value ($r_1$) is called the "fine mode", while the other mode is the "coarse mode." Therefore the relative sizes of the aerosol modes is described by the "coarse mode fraction" $f_c = N_2/(N_1+N_2)$. (Changes in the absolute values of $N_1$ and $N_2$ alter the magnitude of the $AE$ for a given distribution, but do not change the
shape of the ASD for a given $f_c$ value.)

The ASDs used in several other LS $AE$ retrieval algorithms are given in Table 2. These properties have typically been taken from the long record of balloon-borne optical particle counter (OPC) data provided by T. Deshler's group at U. of Wyoming. But this data set indicates that the ASD varies considerably with time, location, and altitude. For example, the V1.1 SCIAMACHY ASD (Von Savigny et al., 2015) is taken from Fig. 3c in Deshler et al. (2008) (excluding the coarse mode).
Bourassa et al. (2007) and Rieger et al. (2014) cite Deshler et al. (2003) as the source of the V5 OSIRIS ASD, which resembles Fig. 5b of that reference (again excluding coarse mode particles). Nyaku (2016) uses the 2012-2013 data provided by the U. of Wyoming web site for Laramie as the basis of the bi-modal LN ASD for sensitivity studies, as cited earlier in Loughman et al. (2015). Unfortunately, the OPC data corrections described by Kovilakam and Deshler (2015) occurred after the OSIRIS, SCIAMACHY and Nyaku ASDs described in this paragraph were defined, so none of those ASDs reflect the corrected version
of the OPC data.

The apparent lack of consistency in the stratospheric aerosol ASD poses a significant problem for efforts to retrieve $AE$ from LS measurements, as discussed further in Sect. 5.2. A single-mode LN ASD is assumed in stratospheric $AE$ retrievals by the V5 OSIRIS (Bourassa et al., 2007), V1.1 SCIAMACHY (Von Savigny et al., 2015), and the intermediate V0.5 OMPS LP retrievals, as shown in Table 2. The assumed mode radius ($r_0$), mode width ($\sigma$) and the resulting Angstrom coefficient
$\alpha(525/1020)$ (defined below in equation ( 2)) are shown in Table 2, and several single-mode and bi-modal LN ASDs are shown in Fig. 7. Table 2 also includes the properties of the bi-modal LN ASD analyzed by Nyaku (2016).

$$\alpha(525/1020) = \frac{-\log\left[AE(525nm)/AE(1020nm)\right]}{\log\left[525/1020\right]} \tag{2}$$

For the V1 OMPS LP $AE$ retrieval algorithm, we introduce the added complexity of the bi-modal LN ASD because it generally describes the properties of stratospheric aerosol observations better (Thomason and Peter, 2006). The fine and coarse
mode properties of the V1 OMPS ASD (given in Table 1) were selected based on the data found in Table 1a of Pueschel et





al. (1994). These observations were taken on Aug. 23, 1991, in the aftermath of the eruption of Mt. Pinatubo, and are based on in situ measurements by impactor samplers flown on an ER-2 aircraft in the lower stratosphere. The intention of this choice was to keep the observed "fine mode" for stratospheric aerosols (with properties broadly similar to the single-mode LN ASDs shown in Table 2), while introducing the possibility of a "coarse mode" of larger aerosols. The recent eruption of Mt. Pinatubo

causes $f_c = 0.36$ in the selected Pueschel et al. (1994) data, which is much larger than one would expect in the background stratosphere. Therefore the relative prominence of the coarse mode was reduced for the V1 OMPS LP $AE$ algorithm by tuning the $f_c$ value, based on the following considerations drawn from the available stratospheric aerosol data record:

1. The SAGE satellite series (particularly SAGE II) provides a long-term record of $AE$ profiles for stratospheric aerosols at several wavelengths. The $AE$ wavelength variation can be expressed by the Angstrom coefficient $\alpha$, which is defined by

equation ( 2) based on observations of $AE$ at 525 and 1020 nm. The SAGE II zonal mean $\alpha$ value for the tropics at 30 km is shown in Fig. 8. Except for volcanically-perturbed periods, the observed $\alpha$ value is relatively constant at $\alpha \approx 2$.

2. Fig. 9 shows how $\alpha$ varies with coarse mode fraction $f_c$, for fine and coarse mode fraction values in the vicinity of the V1 OMPS LP ASD values ($r_1, \sigma_1, r_2, \sigma_2$ in Table 1). For these assumed fine and coarse mode properties, the value of $\alpha$ is extremely sensitive to $f_c$. If one assumes that the fine and coarse modes are correctly specified, this implies that $f_c$ can be

determined with great precision based on the observed value of $\alpha$. The V1 OMPS LP $AE$ retrieval algorithm uses $f_c = 0.003$ in conjunction with the Pueschel et al. (1994) values of $(r_1, \sigma_1, r_2, \sigma_2)$ to produce $\alpha = 2$.

The differences among the V1 algorithm assumed $APF$ and the phase functions associated with other LS $AE$ retrievals are shown in Fig. 10, and discussed further in Sect. 5.2.

### 3.4 Properties of Altitude-Normalized Radiances (ANR)

As explained in Sect. 4.1, the V1 algorithm uses altitude-normalized radiances ($ANR$) rather than radiances to define the measurement vector $y$. The $ANR$ is defined as $\rho = I(\lambda, h)/I(\lambda, h_n)$, with the radiance at the tangent height $h$ of interest divided by the radiance at a selected normalization tangent height $h_n > h$. For the V1 algorithm, $h_n = 40.5$ km. In Fig. 11, the $ANR$ at 675 nm is calculated for a range of scattering angles using the V1 OMPS LP ASD. The $AE$, ozone, pressure and temperature profiles are fixed for the radiance calculations shown in Fig. 11, in order to isolate the dependency of $ANR$ on

$SSA$ and $R$.

When aerosols are excluded from the model atmosphere, Fig. 11 shows that the $ANR$ is insensitive to both $SSA$ and $R$. But when aerosols are included, several effects emerge:

1. $ANR$ is sensitive to $SSA$ due to the strong variation of the $APF$ with $SSA$, as shown in Fig. 5. For cases in which $R$ is low, the variation of $ANR$ with $SSA$ can be estimated by the variation of the phase function ratio $APF/RPF$, in which the

$APF$ is divided by the Rayleigh phase function $RPF$. The phase function ratio varies with $SSA$ as shown in Fig. 12.

2. $ANR$ also shows some dependence on $R$ when aerosols are included. However, this effect is relatively small compared to the effect of $R$ on the radiance, which can reach $100\%$ at large values of $R$.

3. The correlation of $ANR$ with $APF/RPF$ is also reduced somewhat as $R$ increases. As the underlying scene becomes brighter, the limb radiance is influenced more by $DUR$. This upwelling radiation illuminates the LOS from a variety of



directions, reducing the influence of the solar scattering angle $SSA$ on the $ANR$. As a result, the $ANR$ becomes less sensitive to the details of $APF(SSA)$ as $R$ increases.

## 4 Retrieval Algorithm

### 4.1 Aerosol Scattering Index ($ASI$)

The V1 algorithm uses the Aerosol scattering Index ($ASI$) as its measurement vector $y$. The $ASI$ is defined as $y(\lambda, h) = (\rho_m - \rho_R)/\rho_R$, where $\rho_m$ is the measured $ANR$, and $\rho_R$ is the $ANR$ calculated assuming an aerosol-free (and therefore purely Rayleigh-scattering) atmosphere bounded by a Lambertian reflecting surface of reflectivity $R$. The value of $R$ is derived from 675 nm sun-normalized radiances measured at $h_n = 40.5$ km, as discussed in Sect. 3.4. The radiance calculation that determines $R$ assumes that no aerosols are present along the LOS at $h_n = 40.5$ km, which forces $ASI = 0$ at $h_n$. We initially

assume a climatological ozone profile to account for the weak ozone absorption at 675 nm. The ozone estimate is then updated at the final step of the retrieval, as described in Sect. 4.3.

   For an optically-thin LOS, we can use the SS approximation and treat the $ASI$ as a sum of $\rho_a$ (the $ANR$ due to aerosol scattering) + $\rho_R$ (the $ANR$ due to Rayleigh scattering). In that case, the measured $ASI = \rho_m \approx \rho_a + \rho_R$, and therefore the $ASI \approx \rho_a/\rho_R$. It is also true under these conditions that $\rho_a \approx AE \times APF$. However, under more general conditions the

scattering contributions cannot be treated independently: Attenuation of Rayleigh scattered photons by aerosols can cause the $ASI$ to become negative at some altitudes. This indicates that the aerosol attenuation effect has exceeded the aerosol scattering effect. This behavior can be seen in Fig. 13, particularly at the southern end of the orbit (where the OMPS LP aerosol signal is weakest). Finally, note that the strong hemispheric contrast that exists in the $ASI$ picture (Fig. 13) simply reflects the $APF(SSA)$ contrast, and therefore is not repeated in the corresponding $AE$ picture (Fig. 14).

### 4.2 Inverse Model

The V1 algorithm uses OMPS LP radiance measurements at a single wavelength (675 nm) to estimate the $AE$ profile. This wavelength was selected primarily to provide aerosol information to the V2.5 ozone code that uses a wavelength triplet (consisting of 510, 600, and 675 nm) to retrieve the ozone profile (Kramarova et al., 2017). Since both $AE$ and $APF$ have strong wavelength dependence in the stratosphere, aerosol profiles derived from a wavelength near the Chappuis ozone band are

expected to minimize aerosol-related errors in the ozone retrieval.

   Several additional advantages make selecting a wavelength near 700 nm optimal for OMPS LP aerosol retrievals. Wavelengths < 500 nm feature weak ozone absorption, but large Rayleigh scattering obscures the aerosol signal. OMPS LP also measures wavelengths longer than 675 nm, but these tend to be more affected by internal instrument stray light (SL). The OMPS LP instrument was designed and characterized primarily with the goal of ozone retrieval, and therefore successful

characterization of SL at the longer wavelengths is an ongoing project. Longer wavelengths are also more sensitive to the highly-uncertain ASD than 675 nm, making 675 nm attractive for $AE$ retrievals.





The V1 algorithm uses the Chahine non-linear relaxation method (Chahine, 1970) to obtain the $AE$ from the OMPS LP measurements. Since $ASI$ is roughly proportional to $AE$, we use $ASI$ as the measurement vector $y$, which is updated iteratively as shown in equation ( 3), based on the notation of Rodgers (2000), Sect. 6.8:

$$x_i^{n+1} = x_i^n \frac{y_i^m}{y_i^n} \tag{3}$$

5     The symbol $x_i^n$ represents the state vector ($AE$) at altitude $z_i$ after $n$ iterations of the retrieval algorithm. The measurement vector $y_i^m$ represents the measured $ASI$ at tangent height $h_i = z_i$. The GSLS RT model calculates the $ASI$ vector $y_i^n$ at each iteration, using the $AE$ profile given by $x_i^n$. The iterative process is initialized with a nominal first-guess aerosol profile $x_i^0$ derived from 2000-2004 SAGE data (shown as Fig. 15), which does not vary with latitude or season.

    The retrieval is constrained to limit changes within a single iteration: $x_i$ can increase by no more than a factor of 2, while decreases are limited to be a factor of 5 or less in each iteration. The algorithm executes just 3 iterations, which constrains the final solution at each altitude $x_i^3$ within the range of values $x_i^0/125 \le x_i^3 \le 8x_i^0$. The retrieval algorithm sets $x_i$ to zero for observations with weak aerosol signals (where $y_i^m < 0.01$). Data at altitudes for which a cloud has been detected by the algorithm described by Chen et al. (2016) is flagged.

    For this algorithm the fractional error in $x$ due to error in $y$, called the "Gain" $G$ by (Rodgers, 2000), can be shown to have a particularly simple form, given as equation ( 4):

$$G = \frac{d \log x_i}{dy_i} = \left| \frac{1}{y_i} \right| \tag{4}$$

### 4.3 Ozone Correction

The V1 $AE$ algorithm operates independently from the ozone retrieval algorithm (Kramarova et al., 2017). As noted in Sect. 4.1, a climatological ozone profile is assumed during the iterations of the $AE$ retrieval. After those 3 iterations are complete, an approximate ozone correction is applied as follows. For $\lambda_1, \lambda_2, \lambda_3 = 510, 600, 675$ nm, we define $Y(h, \lambda_i) = Y_i$ as:

$$Y_i = \log \left[ \frac{I_m(h, \lambda_i)}{I_c(h, \lambda_i)} \right] \tag{5}$$

Based on these three $Y$ values, we define a three-parameter fit:

$$Y_i = a + b\lambda_i + c\sigma_i \tag{6}$$





where $\sigma_i$ = the ozone absorption cross-section averaged over the OMPS LP bandpass centered at $\lambda_i$. The $c$ parameter represents the sensitivity of the ozone slant column density with respect to the first guess, and can be determined from equation (7):

$$c = \frac{(Y_2 - Y_1)(\lambda_3 - \lambda_2) - (Y_3 - Y_2)(\lambda_2 - \lambda_1)}{(\sigma_2 - \sigma_1)(\lambda_3 - \lambda_2) - (\sigma_3 - \sigma_2)(\lambda_2 - \lambda_1)} \tag{7}$$

The ozone-corrected value of $Y$ at 675 nm is therefore denoted by $Y_c(\lambda_3)$:

$$Y_c(\lambda_3) = Y(\lambda_3) \exp\left[c\sigma(\lambda_3)\right] \tag{8}$$

A similar correction is also applied to the value of $Y$ at the normalization tangent height to obtain $Y_c(h_n, \lambda_3)$.

## 5   Error Analysis

This section describes the most significant categories of uncertainty that we anticipate will limit the accuracy and precision of

the V1 retrievals. Quantitative estimates of the anticipated error are provided when possible, but a full algorithm error budget is beyond the scope of this study. Unfortunately, many uncertainties are difficult to quantify for the full range of possible conditions.

### 5.1   Uncertainty Due to Measurement Errors

As defined in Sect. 4.1, our measurement vector $y$ is influenced by 4 radiances (all at $\lambda = 675$ nm): The measured radiance

at the tangent height of interest $h_i$ and the normalization tangent height $h_n$, and the calculated radiance (excluding aerosol from the model atmosphere) at the same tangent heights. The primary source of error in $y$ appears to be the stray light (SL) error at $h_n$. OMPS LP stray light acts roughly as an additive effect (Jaross et al., 2014), and therefore affects the measured radiance at $h_n$ much more strongly than the other radiances that form $y$, due to the roughly exponential decrease of $I$ with tangent height. Internal analysis suggests that this error is 1%, and therefore produces fractional error in $x = 0.01/y$. Stray

light error therefore becomes most significant at altitudes and latitudes where the $ASI$ is small ($< 0.1$). As shown in Fig. 13, this condition is most likely to occur near the top of the Junge layer ($h \approx 35 - 40$ km), and/or near the South Pole (where SNPP OPMS LP provides unfavorable viewing conditions for $AE$ retrieval, with large $SSA$ producing small $APF$ values).

### 5.2   Uncertainty Due to Radiative Transfer Limitations

The GSLS radiative transfer model used in the V1 OMPS LP $AE$ retrieval algorithm contains several limitations that affect the

retrieved $AE$ profiles. The most significant issues are listed below, in order of priority.

    1. Uncertainty in the aerosol scattering phase function $APF$





As described in Sect. 3.3.2, we have selected a bi-modal LN ASD to calculate the assumed $APF$ used in the V1 $AE$ retrieval algorithm. However, we cannot expect that any single ASD will be correct for the full range of OMPS LP observations. And even if a single ASD were suitable, many plausible combinations of $r_1, \sigma_1, r_2, \sigma_2$, and $f_c$ exist that would fit the criterion stated in Sect. 3.3.2 ($\alpha \approx 2$) equally well, as shown in Fig. 16. Whether these "plausible" ASDs produce significantly different $APF$

values depends strongly on $SSA$. As shown in Fig. 5, the $APF$ for back-scattered directions varies much more strongly with $SSA$ than the $SSA = 30 - 90°$ directions. The sensitivity of $APF$ to ASD for the cases shown in Fig. 16 are illustrated in Figs. 17 - 18.

Since $\rho_a$ is approximately proportional to $APF$ for optically thin LOS, differences between the assumed and true $APF$ values map directly into $AE$ errors in the V1 algorithm. Fig. 18 therefore predicts that the OMPS LP $AE$ retrievals for

$SSA = 120°$ will be greatly affected by the assumed ASD in the retrieval, while Fig. 17 shows that the OMPS LP $AE$ retrievals for $SSA = 60°$ will be nearly insensitive to the assumed ASD. The preceding analysis roughly estimates the possible error that may result in the V1 OMPS LP $AE$ retrievals, but no clear method to estimate the error in a single retrieval at a particular place, time and altitude. This topic will be explored more thoroughly in a future publication, but Fig. 19 allows one to estimate the sensitivity to various perturbations from the baseline V1 OMPS LP ASD.

2. Uncertainty due to LOS variation in atmospheric properties

As noted in Sect. 3.1, the RTM in the V1 OMPS LP $AE$ retrieval assumes that the atmospheric properties vary only with altitude. This assumption is used to retrieve $AE$ for each measured image, independent of the neighboring images. But the maps of retrieved $AE$ values regularly feature large horizontal variations, particularly latitudinal variations (see Fig. 14). Many such features persist at particular latitude ranges for which stratospheric dynamics are known to cause steep horizontal

gradients in $AE$ at a given altitude.

The viewing geometry of OMPS LP (looking backwards along the sun-synchronous orbital track) exacerbates this problem, due to the zonal gradients in $AE$ seen in Fig. 14, but LOS variations of atmospheric properties affect all limb-viewing retrieval methods. Past limb missions have developed a two-dimensional retrieval strategy that allows variation of the retrieved quantity both along the LOS and with altitude. The MLS (limb emission) mission (Livesey and Read, 2000) and OSIRIS (LS) mission

(Zawada et al., 2015) have made notable progress in this area. The V1 OMPS LP algorithm remains a 1D solution (with $AE$ varying only with altitude). This assumption is likely to affect the retrieval most strongly at the edge of the tropics (where $AE$ tends to have a large horizontal gradient), in the Northern Hemisphere (where $ASI$ varies rapidly with $SSA$), and at the edges of a fresh volcanic cloud.

3. Uncertainty due to approximate treatment of $DUR$

The limb LOS is illuminated from above (overwhelmingly by direct solar radiation) and from below (by photons scattered within the underlying atmosphere and/or reflected by the underlying surface). The latter source of radiation is modeled as described in Sect. 3.2: A Lambertian surface is assumed to lie beneath the model atmosphere (which is not updated outside the range at which the $AE$ is retrieved during the iteration process). This assumption allows one to determine $R$, the effective Lambertian surface reflectivity that is consistent with the measured radiance at $h_n = 40.5$ km.



This assumption provides a first-order estimate of the $DUR$, but this estimate will generally be imperfect for the following reasons:

    a. The simple assumptions described above generally fail to represent the true conditions below a given LOS in multitple ways: The atmosphere will generally include clouds and aerosols below the LOS that are not included in the model atmosphere.

The true BRDF of the scene will also generally be non-Lambertian. In such cases, the upwelling radiation in the model calculation will have a different angular distribution than the upwelling radiation in the true atmosphere.

    b. For an inhomogeneous underlying scene, the effective $LER$ may also vary with $h$, due to the varying solid angle that contributes to $I(h)$. The difference between $LER$ ($h = 40$ km) and $LER$ ($h = 50$ km) is typically slight (see Fig. 20), implying that this is a minor effect, but more research is needed to assess whether any systematic relationships exist.

## 5.3 Inverse Model Errors

This section includes several effects unrelated to the radiative transfer model that affect the V1 OMPS LP $AE$ retrieval, again listed in order of priority.

    1. Large aerosol extinction

    As noted in Sect. 4.2, the algorithm limits possible variation of the retrieved $AE$ value. As a result, the retrieval often "sat-

urates" at the maximum allowed value when the $AE$ is large relative to the the first-guess profile. At higher extinction values, the retrieval will also be more influenced by inhomogeneity along the LOS, since the LS radiance will be more influenced by the LOS segment nearest the sensor (see item 3 below).

    2. Cloud detection algorithm

    The current cloud detection algorithm (Chen et al., 2016) detects clouds well, but it sometimes also flags fresh volcanic

aerosols as clouds. Since retrieval of such aerosols is quite complicated for several reasons discussed earlier (LOS inhomo- geneity, uncertainty about the appropriate $APF$ due to a mixture of aerosol types and shapes, etc.), we have not attempted to fix this error.

    3. Poor convergence

    The algorithm often doesn't converge well for scenes in which the $ASI$ has large horizontal gradient. We believe that this

occurs because of 2D effects discussed earlier in Sect. 5.2, which produce an asymmetry in the LS radiance contribution function. Under optically thick conditions, the LS radiance will be influenced by the atmospheric properties at a given altitude near the satellite much more than the atmosphere the same altitude in the more distant portion of the LOS. This effect is illustrated in Fig. 6c of Loughman et al. (2015). Fixing this problem will require the development of a 2D aerosol algorithm.

## 5.4 Ozone Correction Errors

The 675 nm radiances used in the V1 OMPS LP $AE$ retrieval algorithm lie within the Chappuis ozone absorption band, and therefore the $AE$ estimate is influenced by possible differences between the true ozone profile and the ozone profile that is assumed in the calculation of $y_i^n$ in equation ( 3). We therefore apply the ozone correction described in Sect. 4.3 to reduce





this source of error. This correction produces the largest percentage change in the retrieved $AE$ value when the following conditions are met:

1. The a-priori ozone concentration differs signficantly from the true ozone concentration.

2. The $ASI$ is relatively small for a given $AE$ value.

3. The $AE$ value itself is small.

The first condition is most likely to occur for regions with highly variable ozone profiles. The second condition will prevail for regions that are viewed by OMPS LP at large $SSA$ values, where the corresponding $APF$ value is small. The third condition occurs primariily in regions with low $AE$ values, typically where sinking air prevails in the UT/LS region.

The largest ozone corrections therefore typically appear near the South Pole, where minima for both the $ASI$ and $AE$ at a

given altitude tend to occur, as shown in Figs. 13 and 14, respectively. The ozone profile also exhibits large variation in this region, partly due to the formation of the Antarctic spring ozone hole. Under these extreme conditions, the ozone correction produces changes in the retrieved $AE$ value as large as $20\%$. For a more typical case in the tropics, the $AE$ changes by $< 3\%$ when the ozone correction is applied.

## 6    Preliminary Evaluation of Retrieval Results

In this section, we will only present an early qualitative evaluation of OMPS LP V1 $AE$ data in comparison with profiles derived from OSIRIS LS radiances and CALIPSO (Winker et al., 2009) backscattered LIDAR measurements. A detailed validation paper for the OMPS LP $AE$ retrievals is in preparation.

Fig. 21 shows OMPS LP V1 and OSIRIS V5.07 retrieved $AE$ in the tropics. In general, the two data sets agree to within $25\%$. OSIRIS daily means are noisier because of its relatively limited coverage, which provides fewer profiles for a given day

compared to OMPS. Both OMPS and OSIRIS show enhanced aerosol values at 18.5 km and 20.5 km following the tropical volcanic eruptions of Nabro (June 2011) and Kelut (February 2014). Transport of the plume associated with Calbuco (which erupted in the southern hemisphere in May 2015) is also evident. At 20.5 km, OMPS measurements are lower than OSIRIS during the peak of Kelut plume, most likely caused by the retrieval's restriction on the number of iterations (see Sects. 4.2 and 5.3), although differences between the OMPS LP and OSIRIS coverage patterns can contribute to such differences. At 30.5

km, both instruments clearly show the quasi-biennial oscillation (QBO) signature of enhanced $AE$ values during easterly shear conditions of the QBO (Trepte and Hitchman, 1992) during early 2012, 2013-2014, 2016, caused by enhanced aerosol lofting. The lower values of $AE$ in 2012 and 2015 are associated with westerly shear conditions of the QBO, causing downward aerosol transport.

Fig. 22 shows monthly zonal mean $AE$ profiles at 750 nm derived from CALIPSO, OSIRIS and OMPS LP measurements

during 2014. This time series is averaged from $5°$ S to $0°$ S, and altitudes 15-35 km are illustrated. CALIPSO data was provided by Vernier et al. (2011) and Vernier et al. (2015). The three instruments track Kelut injection of volcanic aerosol at 20 km and the upward lofting of the aerosol to higher altitudes ($\approx 25$ km) within a few months. The CALIPSO data is based on a series





of narrow LIDAR swaths, so its coverage differs from OSIRIS and OMPS LP coverage. Vertical resolution differences might also explain some the differences seen among the 3 instruments.

## 7   Conclusions

The OMPS LP V1 aerosol extinction ($AE$) retrieval algorithm is summarized in this document. The V1 algorithm differs from
the most recently-published OMPS LP algorithm (given in  Rault and Loughman (2013)) in several ways:

    1. The $AE$ profile is retrieved at a single wavelength, 675 nm.

    2. The retrieval uses the  Chahine (1970) solution method.

    3. The assumed ASD is bi-modal log-normal, guided by the aerosol properties measured by Pueschel et al. (1994) with the
coarse mode fraction tuned to produce Angstrom coefficient $\alpha(525/1020) \approx 2$.

10       The main motivation for these changes was to produce a simpler algorithm that works with the best-characterized OMPS LP radiances. The resulting $AE$ profiles are more stable, and permit more straightforward analysis of the radiance residuals. Initial comparisons with coincident OSIRIS and CALIPSO $AE$ data show similar spatial and temporal variation over the lifetime of the OMPS LP instruments.

    The accuracy of the absolute value of the OMPS LP $AE$ remains variable, primarily due to uncertainty about the appropriate
ASD to be used. The V1 ASD selection was guided by the Angstrom coefficient measured by SAGE II during volcanically quiescent periods. But the lack of contemporaneous global observations of the ASD presents a significant challenge for all LS $AE$ retrievals, particularly for observations at $SSA > 90°$ (Southern Hemisphere conditions for OMPS LP). The recently-launched ISS SAGE III instrument is capable of both SO and LS observations, which should provide valuable information to reduce uncertainty in the $APF$ for stratospheric aerosols.

20       Future work to improve the OMPS LP $AE$ algorithm will begin by adding consideration of additional wavelengths. Longer wavelengths are sensitive to lower tangent heights that typically saturate at 675 nm due to interference by Rayleigh scattering, and are also more sensitive to small aerosol signals (such as OMPS LP encounters in the Southern Hemisphere). Additional wavelengths also will allow us to asses the self-consistency of the measured $AE$ wavelength variation with the Mie theory prediction for the assumed ASD. A 2D algorithm will also improve performance in the vicinity of large horizontal variations.
The ability to allow the ASD to vary with height will also be valuable, given better ASD information.

*Acknowledgements.*  This paper includes material that first appeared at the 8th and 9th Atmospheric Limb Workshops, which were hosted by Chalmers University (Gothenburg, Sweden, September 2015) and the University of Saskatchewan (Saskatchewan, Canada, June 2017), respectively. This research was supported by NASA Goddard Space Flight Center through SSAI Subcontracts 21205-12-043 and 21702-17-010. The authors recognize the contributions of the SAGE, OSIRIS, SCIAMACHY, CALIPSO, and U. of Wyoming teams to maintaining
high-quality stratospheric aerosol data, and particularly thank Larry Thomason, Terry Deshler, Adam Bourassa, Landon Rieger, Christian von Savigny, Alexei Rozanov, and Jean-Paul Vernier for helpful insights into the stratospheric aerosol problem. Surendra Bhatta contributed significantly to preparing the figures and references. The NASA, SSAI and NOAA OMPS teams supported this research and contributed many



useful discussions, including Larry Flynn, Matt DeLand, Jack Larsen, and Tong Zhu. Finally, several summer research students contributed to studies that have improved this work, including Nelson Ojeda, Ryan McCabe, and Ashley Orehek.



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





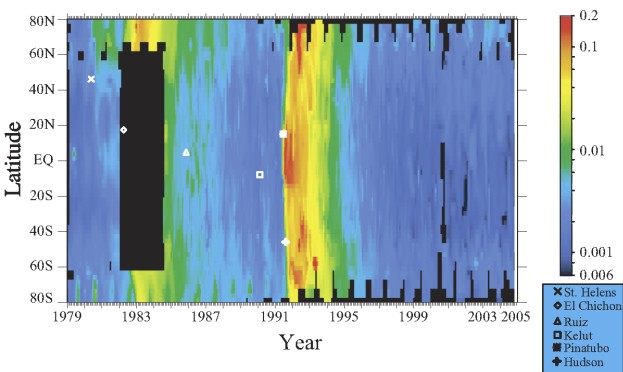

**Figure 1.** The 1020-nm stratospheric optical depth from SAM II, SAGE, SAGE II and SAGE III for the period from January 1979 through the end of 2004. Between the June 1991 Pinatubo eruption and mid 1993, $AE$ profiles are supplemented by lidar data following the method described in (Thomason and Peter, 2006). (From Thomason et al. (2008))

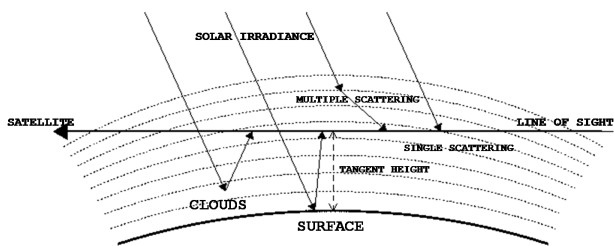

**Figure 2.** Illustration of the various photon paths possible in the LS viewing geometry. (From (Rault and Loughman, 2013)

Zawada, D. J., S. R. Dueck, L. A. Rieger, A. E. Bourassa, N. D. Lloyd and D. A. Degenstein, High-resolution and Monte Carlo additions to the SASKTRAN radiative transfer model , Atmos. Meas. Tech., **8**, 2609-2623, https://doi.org/10.5194/amt-8-2609-2015, 2015.



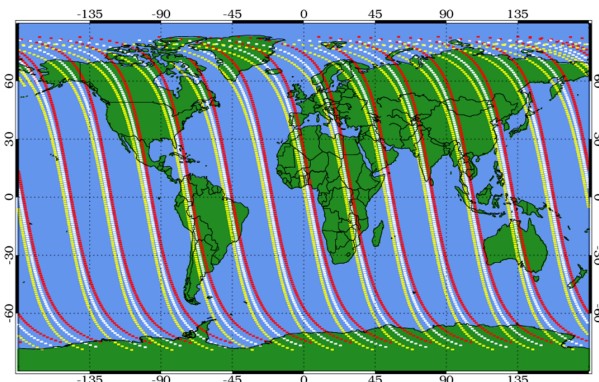

**Figure 3.** Daily coverage provided by the OMPS LP instrument mounted on the SNPP satellite. The tangent point for the LOS corresponding to each observation is indicated, with red, white and yellow circles depicting the left, center and right slit observations.

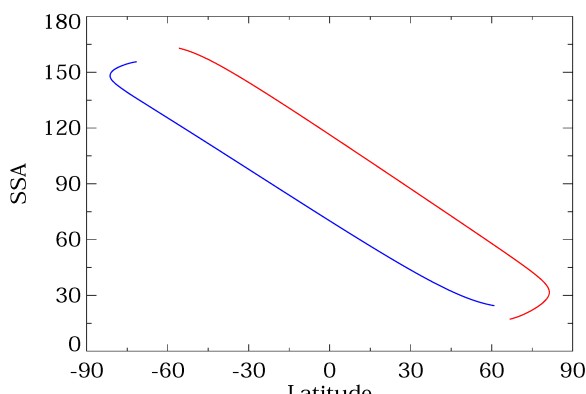

**Figure 4.** The single scattering angle ($SSA$) as a function of latitude for the SNPP OMPS LP instrument. June and December solstice conditions are illustrated by the red and blue lines, respectively. Note that near-polar latitudes may be observed twice (during the ascending and descending nodes of the orbit), which provides useful diagnostic information.





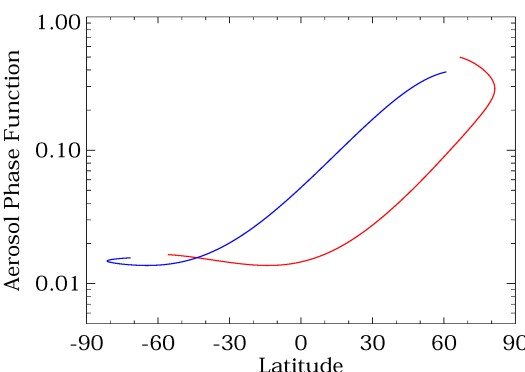

**Figure 5.** The $APF$ (for the $SSA$ values shown in Fig. 4) as a function of latitude for the SNPP OMPS LP instrument. June and December solstice conditions are illustrated by the red and blue lines, respectively. Due to the variation of $APF$ with latitude and season, the SNPP OMPS LP observations are most sensitive to aerosols in the NH winter, and least sensitive in the SH. The aerosol size distribution described in Table 1 for the V1 $AE$ algorithm is assumed.

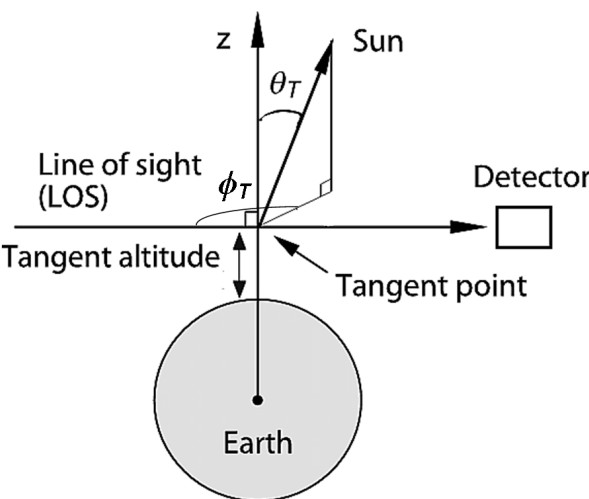

**Figure 6.** Illustration of the limb scattering viewing geometry, including definitions of the tangent altitude $h$ and tangent point. The solar zenith angle and solar azimuth angle at the tangent point are indicated by $\theta_T$ and $\phi_T$, respectively. Adapted from Fig. 1 of (Griffioen and Oikarinen, 2000). Note that a frequently-committed error in the definition of $\phi_T$ (Griffioen and Oikarinen, 2000; Loughman et al., 2004; Bourassa et al., 2008b) has been corrected: A beam with $SSA = 0°$ (scattered exactly forward) has $\phi_T = 0°$.





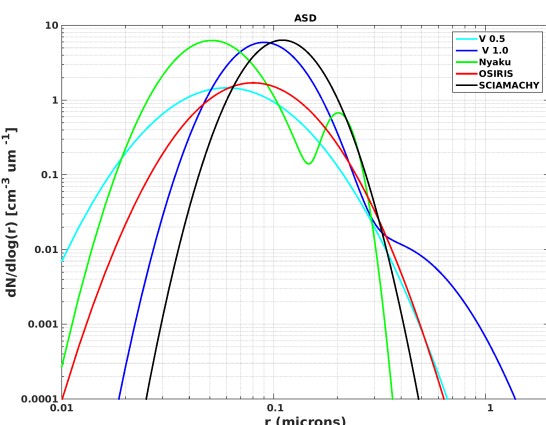

**Figure 7.** Illustration of the ASD used in several recent LS $AE$ retrieval algorithms, including OSIRIS (V5), SCIAMACHY (V1.1), and OMPS V0.5 and V1. The ASD studied by (Nyaku, 2016) is also represented.

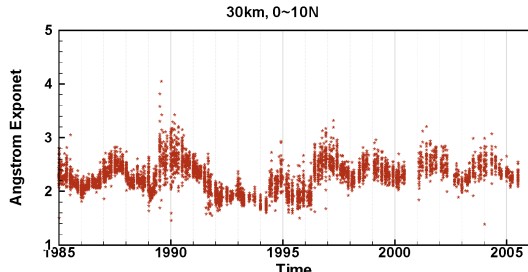

**Figure 8.** Angstrom exponent $\alpha(525/1020)$ derived from SAGE II SO measurements during its measurement history. This picture corresponds to measurements at altitude 30 km for the $0 - 10°$ North latitude bin. Cases for which the measured $AE$ at 1020 nm $< 4 \times 10^{-6}$ km were excluded from this analysis [L. Thomason, private communication].





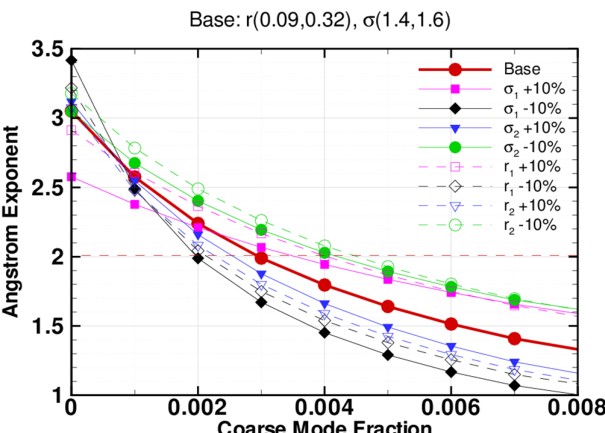

**Figure 9.** Variation of Angstrom exponent $\alpha(525/1020)$ with aerosol properties for the V1 OMPS LP $AE$ retrieval algorithm characteristics. Each curve shows the variation of $(\alpha(525/1020)$ with $f_c$ for a given set of mode radii and mode widths. In addition to the "base" curve (which uses the V1 characteristics listed in Table 1), several curves show how the value of $\alpha(525/1020)$ changes as the values of $(r_1, \sigma_1, r_2, \sigma_2$ in Table 1) are perturbed by $\pm 10\%$.

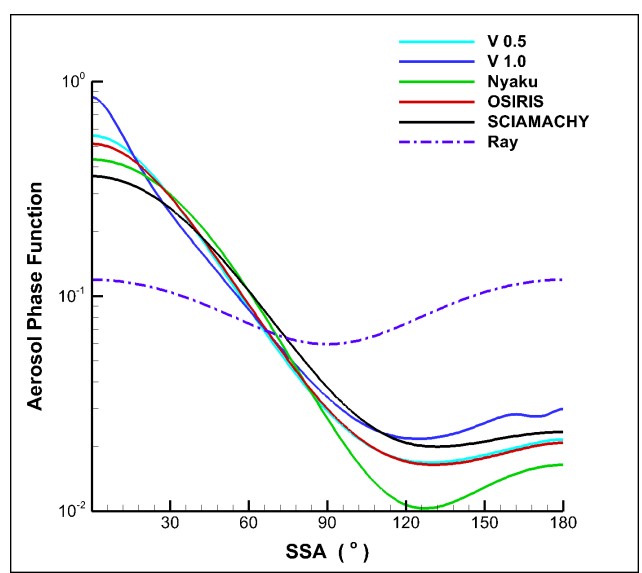

**Figure 10.** The $APF$ as a function of $SSA$ for the ASDs listed in Tables 1 - 2.



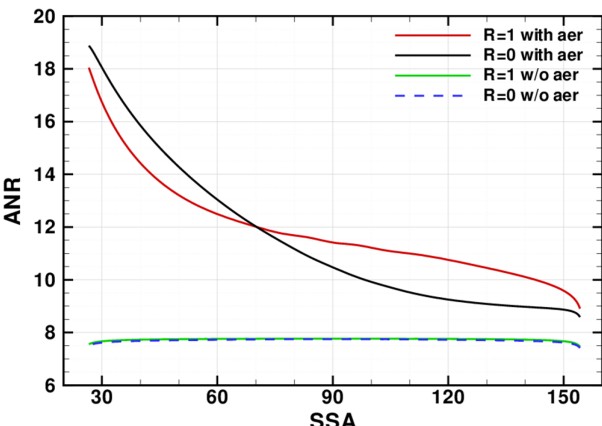

**Figure 11.** ANR(675 nm) as a function of SSA under aerosol-free and aerosol conditions, with both non-reflecting ($R = 0$) and perfectly-reflecting ($R = 1$) surface conditions.

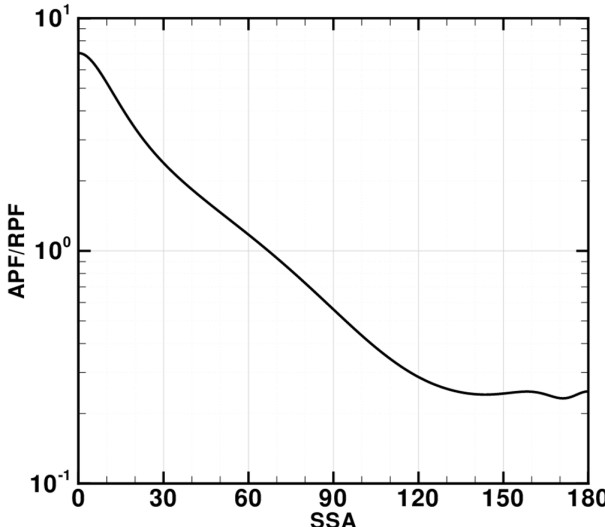

**Figure 12.** The ratio of $APF$ to $RPF$ for the V1 OMPS LP ASD. This ratio declines by a factor of $\approx 50$ between forward ($SSA = 0°$) and backward ($SSA = 180°$) scattering conditions.




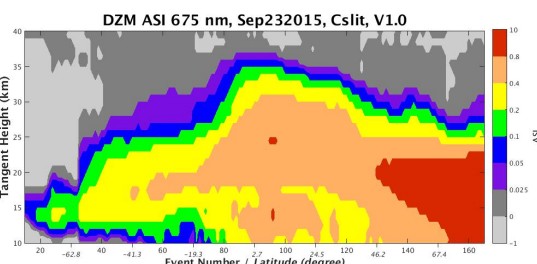

**Figure 13.** Daily zonal mean $ASI$ measured by the SNPP OMPS LP instrument. This picture corresponds to center slit observations on September 23, 2015. The x-axis is labeled with both the event number (solid) and tangent point latitude (italics). The color scale is non-linear, designed to highlight relatively small $ASI$ values in the SH.

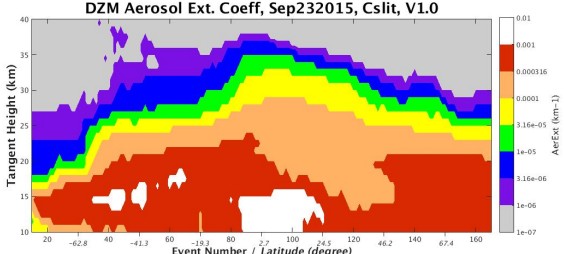

**Figure 14.** Daily zonal mean $AE$ for center slit observations on September 23, 2015 (derived from the $ASI$ measurements shown in Fig. 13).



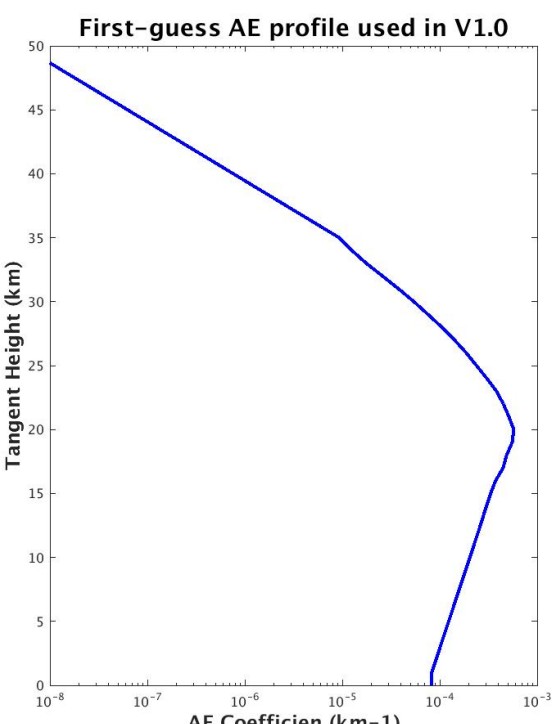

**Figure 15.** The first-guess $AE$ profile used in the V1 OMPS LP $AE$ retrieval algorithm.




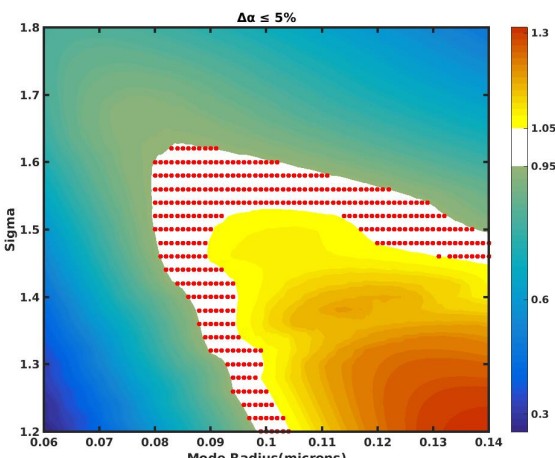

**Figure 16.** Contour plot showing the ratio of the Angstrom coefficient $\alpha$ for a given $ASD$ to the V1 $ASD$ $\alpha \approx 2$. Cases for which this ratio is within $\pm 5\%$ of 1 are highlighted in white. The coarse mode properties are fixed in this example at the V1 ASD values ($r_2 = 0.32\mu$ m, $\sigma_2 = 1.6$), while the fine mode properties vary in the vicinity of the V1 ASD values ($r_1 = 0.09\mu$ m, $\sigma_1 = 1.4$). Red circles indicate the individual cases calculated to create this figure.





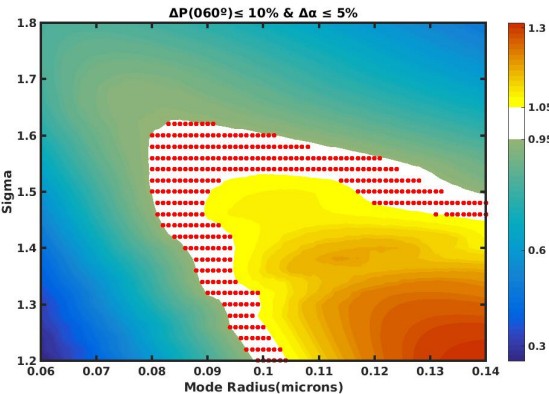

**Figure 17.** The background contour plot is the same as in Fig. 16. This time, red circles appear only for cases in which the Angstrom coefficient ratio is within $\pm 5\%$ of 1 **and** the $APF$ is within $\pm 10\%$ of the V1 $ASD$ value at $SSA = 60°$. Nearly every $ASD$ that satisfies the Angstrom coefficient ratio criterion also satisfies the $APF$ criterion for this case.

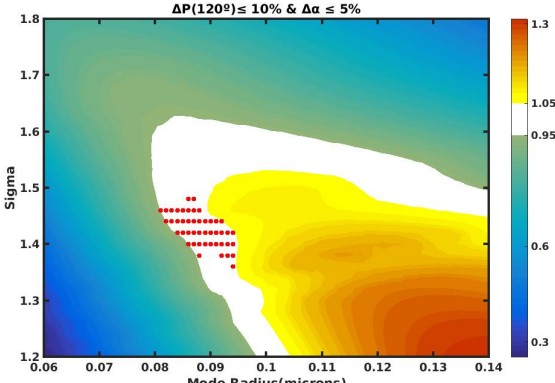

**Figure 18.** Identical to Fig. 17, except that the $APF$ comparison is done for $SSA = 120°$. For this viewing geometry, the $APF$ criterion is much more useful in determining the ASD properties: Note the smaller number of red circles (relative to Fig. 17), centered around the true values of $r_1(0.09\mu\text{m})$ and $\sigma_1(1.4)$.



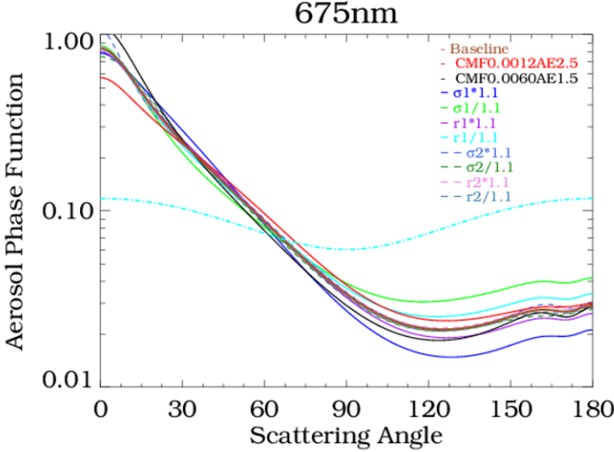

**Figure 19.** $APF$ sensitivity to perturbations of the $ASD$, using the V1 $ASD$ as a baseline. Note that the $\pm10\%$ perturbations of $r_1, \sigma_1, r_2$ and $\sigma_2$ also involved adjustments of $f_c$ to keep $\alpha \approx 2$. The other perturbations simply adjusted $f_c$ to product $\alpha = 1.5$ and 2.5, respectively, without changing the other ASD parameters.

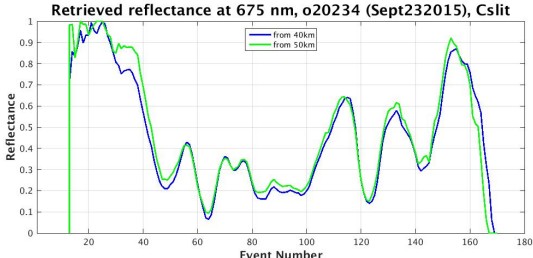

**Figure 20.** $LER$ retrieved from radiances at $h = 40$ km (blue line) and 50 km (green line). Center slit observations from orbit 20234 are used in this example. Again, as noted in Fig. 11, the OMPS LP $AE$ retrieval is insensitive to $LER$.





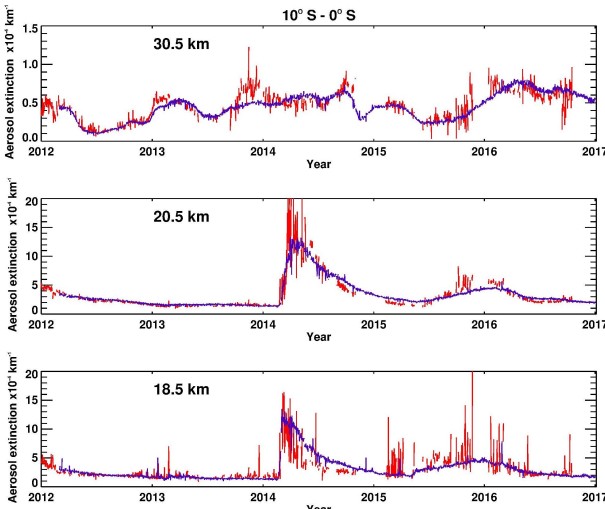

**Figure 21.** OMPS LP V1 (blue line) and OSIRIS V5.07 (red line) retrieved $AE$ daily zonal means at selected altitudes from 2012 to 2016, at latitudes between $10°$ S and $0°$ S. The OSIRIS data set reports $AE$ at 750 nm, so the OMPS $AE$ was converted from 674 nm to 750 nm by using the angstrom coefficient consistent with the ASD assumed in the OMPS LP V1 algorithm.

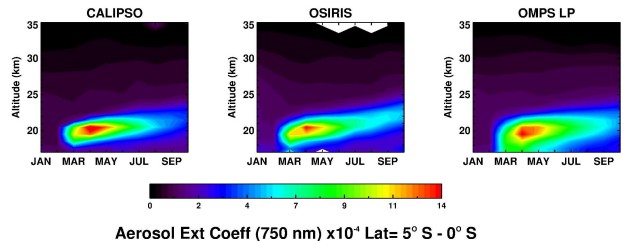

**Figure 22.** Monthly zonal mean $AE$ profiles at 750 nm derived from CALIPSO, OSIRIS and OMPS LP measurements during the aftermath of the Kelut eruption in 2014.



**Table 1.** Aerosol optical properties and aerosol size distribution ($ASD$) assumed in the V1 OMPS LP $AE$ retrieval.

| | |
|---|---:|
| Real aerosol refractive index | 1.448 |
| Imaginary aerosol refractive index | 0 |
| Aerosol mode radius (fine mode), $r_1$ | 0.09 $\mu$ m |
| Aerosol mode width (fine mode), $\sigma_1$ | 1.4 |
| Aerosol mode radius (coarse mode), $r_2$ | 0.32 $\mu$ m |
| Aerosol mode width (coarse mode), $\sigma_2$ | 1.6 |
| Aerosol coarse mode fraction, $f_c$ | 0.003 |
| Aerosol scattering cross-section (at 675 nm) | $1.50 \times 10^{-10}$ cm$^2$ |

**Table 2.** $ASD$ assumed in several recent LS $AE$ retrieval algorithms.

| Mission | Source | $r_0(\mu m)$ | $\sigma$ | $\alpha(525/1020)$ |
|---|---|---:|---:|---:|
| OMPS (V0.5) | (Loughman et al., 2015) | 0.06 | 1.73 | 2.34 |
| OSIRIS (V5) | (Bourassa et al., 2007) | 0.08 | 1.6 | 2.44 |
| SCIAMACHY (V1.1) | (Von Savigny et al., 2015) | 0.11 | 1.37 | 2.82 |
| Nyaku | (Nyaku, 2016), fine mode | 0.05105 | 1.43833 | 2.07 |
| | (Nyaku, 2016), coarse mode | 0.2025 | 1.15 | |