# Peer review of "The Ozone Mapping and Profiler Suite (OMPS) Limb Profiler (LP) Version 1 Aerosol Extinction Retrieval Algorithm: Theoretical Basis"

_Atmospheric Measurement Techniques, 2017_

## Short Comment (SC1) · 10 Oct 2017

Dear Robert,

the paper is very interesting, and it is contributing a lot to the stratospheric aerosol science. I just would like to draw your attention to some issues which might help to avoid a possible misinterpretation of the notations you have used in the paper. To my opinion your formula defining the particle size distribution as well as definition of the mode radii might be slightly confusing. First, it should be unambiguously communicated to the reader that the log-normal distribution is defined as probability distribution of a variable whose NATURAL logarithm is normally distributed. As I understand log(x)

[Figure]

is a common notation for the natural logarithm in the US literature, it could however be misinterpreted as a decimal logarithm in some other countries. My suggestion is to use $\ln(x)$ in equation (1) of the paper instead, to avoid a possible confusion. Furthermore, in accordance with several publications, e.g., SPARC, 2006: SPARC Assessment of Stratospheric Aerosol Properties (ASAP). L. Thomason and Th. Peter (Eds.), SPARC Report No. 4, WCRP-124, WMO/TD - No. 1295, available at www.sparc-climate.org/publications/sparc-reports/ (Sec. 1.4.1), a common notations for $r_1$ and $r_2$ in the log-normal particle size distribution (equation (1) of your paper) is the median radii of the fine and coarse modes, respectively, rather than the mode radii as is given in the line 5 on the page 7. The same issue is encountered in the line 24 on the same page, line 22 on the page 3, Table 1 and further in the text. According to the mathematical statistics, the mode radius of the log-normal distribution is obtained as $R_{mod} = r_{med}/\exp(\ln(\sigma)^2)$. See for example in Johnson, Norman L.; Kotz, Samuel; Balakrishnan, N. (1994), "14: Lognormal Distributions", Continuous univariate distributions. Vol. 1, Wiley Series in Probability and Mathematical Statistics: Applied Probability and Statistics (2nd ed.), New York: John Wiley Sons. It would be nice if you could consider the suggested corrections before final publication of the paper.
* * *

---

## Referee Comment (RC1) · Anonymous Referee #2 · 1 Nov 2017

Global information on stratospheric aerosol extinction profiles as provided from limb scattering and occultation measurements are crucial for understanding variations in stratospheric ozone abundance in times of climate change and volcanic activity. A continuous development of retrieval strategies is required to reduce the uncertainty of the measured aerosol extinction. In their work Loughman et al. improve the existing OMPS LP algorithm that retrieves aerosol extinction at 675 nm. The new Version 1 algorithm is shown to agree well with the OSIRIS V5.07 aerosol extinction data set in the equatorial region between 2012 and 2017. In a comprehensive error analysis, common problems of current aerosol extinction retrieval algorithms are identified and well explained.

[Figure]

**Minor Technical Correction**

- I would recommend using fewer abbreviations. While correctly introduced, they sometimes make it hard to read a paragraph.

---

## Referee Comment (RC2) · Anonymous Referee #1 · 3 Nov 2017

This is a useful paper that describes the OMPS LP version 1 aerosol extinction retrieval algorithm in detail. The background and discussion around various issues involved with limb scatter aerosol retrievals is excellent and really provides some focus on systematic issues that are difficult to deal with. In my opinion, there are a few remaining "major" issues that should be considered; however, I think most are easily addressed. Overall the paper is well done and suitable for AMT.

Major comments:

- Abstract: Is there any evidence to suggest that horizontal variation in aerosol extinction is really a primary limitation? The discussion in the error section of the paper is

good at a qualitative level, but there is no way to conclude this is dominant source of error. Is it greater than the precision? If this statement remains, can it be quantitatively estimated using model data of the expected variability and expected impact on the retrieved profile?

- The motivation for choosing a bi-modal size distribution remains unclear. The discussion about the various choices of size distribution in existing retrieval algorithms is helpful and really points out that this is a problem that needs to be addressed by the larger community (or at least form some consensus). However, one concludes (and the authors point out) from this discussion that we are really information poor on this aspect, so, why proceed to choose an even more complex size distribution requiring more unknown parameters? While it might be appealing to choose a bi-modal size distribution to "introduce the possibility of a 'coarse mode' of larger aerosols", the proposed algorithm does not provide any capability to actually use this in a meaningful way (i.e. it's static). Additionally, the ER-2 observations and the SAGE II Angstrom exponent during the post-Pinatubo time period at high altitude (30 km) are probably not the most representative of the conditions observed globally during the OMPS mission. Even so, the authors claim the SAGE II Angstrom exponent is relatively constant outside volcanically perturbed time period with alpha = 2, but it looks like (from Figure 8) the value is higher than that ($\sim$2.5) and comes down to 2.0 in the years immediately following Pinatubo.

- Is the signal to noise sufficient to use only a single point (h = 40.5 km) for the altitude normalization?

- The third conclusion in Section 3.4 relating to the correlation of normalized radiance and phase function ratio is not clear to me. Plot the correlation perhaps?

- Why not use a "color index" like the OSIRIS and SCIAMACHY retrievals? Also, the choice of 675 nm is motivated by the fact that this is the long wavelength normalization of the Chappuis band ozone retrieval. However, this requires the correction of ozone-

related absorption interference in the aerosol retrieval (which the error analysis shows can have an effect of up to 20%). Why not just move slightly further to the red end of the spectrum and avoid this? An aerosol extinction retrieval at 675 nm still requires some type of extrapolation across the Chappuis band for the ozone retrieval. If stray light is really the limiting factor, then a quantitative statement or plot in this regard would be helpful. Is there a reference for the statement regarding the increased sensitivity of ASD to longer wavelengths? (page 9, line 30-31).

- What is the spectral resolution at 675 nm? How is this handled in the forward model?

- Is three iterations of the retrieval sufficient for convergence, especially at low altitudes? Convergence is a tricky thing to pin down for a non-linear relaxation like this, but a statement about how much the retrieval profile typically changes with more iterations would be insightful.

- What about error due to stray light? Is there any knowledge about how well this is corrected in the Level 1 product?

Minor comments:

- Don't use acronyms/abbreviations in the abstract; prolific use throughout (AE, LP, GSLS, APF, ASD, LN, SO, LS) the manuscript makes it hard to read. For example "AE" is not a widely used acronym and it does make the text clumsy in my opinion

- Page 1, line 14: hydrated sulfuric acid

- Page 1, line 18: the spread of volcanic aerosol is also in the vertical direction, although much more slowly.

- Figure 1: Should not introduce instruments for the first time in a figure caption. Maybe better to first reference this figure in the text after the occultation discussion in the next section.

- Page 2, line 6: Not sure what is meant by "time of day" since the aerosol lifetime is so

long. "Time of year" is more applicable.

- Section 1.2, second paragraph: Mention should be made of the increased complexity required for LS in the forward modelling of the radiative transfer, i.e. multiple scattering, compared to occultation

- Page 2, line 14: don't use "=" in a sentence, also page 4, line 17

- Figures 4 and 5: could be interesting and helpful to include an additional panel of solar scattering angle and aerosol phase function variation over the course of a year at various latitudes, since this will map to the seasonal biases in the retrievals

- Page 6: Strange to call out Fig 11 before Figs 6-10

- What tangent altitude is used for the calculations in Fig 11?

- Page 9, lines 18-19: it would be better to point out this difference in Figs 13 and 14 after the retrieval algorithm is explained.

- Equation 4 makes an assumption about the averaging kernel.

- Figure 19 is out of place and should be discussed.

---

## Author Response (AR1)

We appreciate the suggestions provided in this comment, and have made the following changes in response:

1. All uses of the mathematical symbol "log" have been replaced by "ln".

2. All references to $r_i$ as a "mode radius" have been replaced by reference to "median radius".

[Figure]

[Figure]

We appreciate the suggestion to reduce the number of abbreviations used in the text (made by both reviewers). We have removed some uses of abbreviations (particularly in the figure captions). But our main response hopefully makes the text easier to read without introducing repetitive use of long phrases (aerosol extinction, aerosol phase function, etc.), by replacing acronyms and abbreviations with mathematical symbols. These include the following:

$\beta_a$ for "aerosol extinction" (AE)

[Figure]

$\Theta$ for "solar scattering angle" (SSA)

$P_a$ for "aerosol phase function" (APF)

$P_R$ for "Rayleigh phase function" (RPF)

$\rho$ for "altitude normalized radiance" (ANR)

$y$ for "aerosol scattering index" (ASI)
* * *
[Figure]

Atmos. Meas. Tech. Discuss.,
doi:10.5194/amt-2017-299-AC3, 2017

[Figure]

We appreciate the thoughtful comments presented, and have responded to as many of them as possible (significantly improving the presentation, in our opinion). For the cases in which we do not respond as requested, our motivation arises from the following viewpoint:

As its title states, the primary purpose of this paper is to document the theoretical basis for the Version 1 OMPS LP aerosol extinction retrieval algorithm. The algorithm described in the text was used to create the Version 1 dataset, which was released many months ago, and should be accompanied by a clear explanation. So we have

used the reviewers' comments to improve that explanation as much as possible, but requests to try new analysis approaches, etc. are beyond the scope of this document.

However, we are pleased to continue the general discussion about the various ways that limb scattering measurements might be used to characterize stratospheric aerosol properties. We definitely do not claim that we have "perfected" the best approach to this problem in the Version 1 algorithm. Like the other groups engaged in this effort, we continue to experiment with the algorithm, and look forward to publishing the resulting analysis in future papers (alluding to a few ongoing research efforts in this text).

Point-by-point responses are numbered in the same order as they were given in the review, beginning with the major comments:

1. (Horizontal variation question.) We agree that we have not proven that line of sight variation in aerosol extinction is a major error source. Therefore we have reworded the abstract to present this as an area of concern that warrants further study, rather than a clearly-quantified error source.

2. (Bi-modal size distribution question.) As noted in Sect. 3.3.2, our main motivation for using a bi-modal size distribution arose from the existing OPC dataset, which generally features a bi-modal size distribution at the altitudes where the stratospheric aerosol extinction is greatest. But the problem of how to specify this more complex distribution is a serious concern. Our initial hope was that requiring the resulting Angstrom exponent to = 2 would minimize the importance of the 5 size parameter settings, but that is unfortunately not true in all cases. Given that the sparse OPC dataset (for example) shows clear variation of aerosol properties with time and all 3 spatial dimensions, we suspect that a general consensus on the "best" static size distribution to use for the limb scattering aerosol extinction retrieval application will never be reached.

(Angstrom exponent question) We agree that we chose a poor example from the SAGE II data record to support the claim that the Angstrom exponent should = 2. We therefore replaced Fig. 9 with a sample at a lower altitude (20 km, rather than 30 km), which lies

much closer to the typical peak of the stratospheric aerosol layer, and which shows Angstrom exponent = 2 for the post-Pinatubo period.

3. (SNR question) The SNR at 40.5 km is typically between 550-800, which justifies its exclusion as a significant error source (see the early paragraphs describing an "error floor" that were added to Sect. 5).

4. (Phase function ratio question) The word "correlation" was not well-chosen, and this section has been re-worded to refer to similar functional forms of the phase function ratio and the ANR as they vary with scattering angle.

5. (Choice of 675 nm wavelength question) At the time that 675 nm was chosen, the quality of the stray light correction for OMPS LP wavelengths beyond that point was uncertain (since those wavelengths played no role in the ozone retrieval). We plan to use longer wavelengths in future work.

(Sensitivity of ASD at longer wavelengths question) We have added aerosol phase function figures for the most promising longer wavelengths that OMPS LP can measure (guided by the SAGE III aerosol channels) at 756, 869 and 1020 nm, as Figs. 12-14. The main effect of using longer wavelengths for that set of ASDs is that the agreement for scattering angle = 30 - 90 deg degrades.

6. (Spectral resolution question) OMPS LP resolution at 675 nm is 15 nm, which has now been added to the text.

7. (Convergence question) Allowing the algorithm to do additional iterations rarely causes the residuals to shrink significantly (i.e., the changes are generally below the $1 - 2\%$ "noise floor" discussed in Sect. 5).

8. (Stray light question) As also now noted in Sect. 5, our analysis indicates that the residual stray light error at 675 nm generally is at the $1\%$ level or below.

Minor comments:

[Figure]

All of these have been accepted. Some detailed responses appear below:

1. (Acronym question) We appreciate the suggestion to reduce the number of abbreviations used in the text (made by both reviewers). We have removed some uses of abbreviations (particularly in the figure captions). But our main response hopefully makes the text easier to read without introducing repetitive use of long phrases (aerosol extinction, aerosol phase function, etc.), by replacing acronyms and abbreviations with mathematical symbols. These include the following:

$\beta_a$ for "aerosol extinction" (AE)

$\Theta$ for "solar scattering angle" (SSA)

$P_a$ for "aerosol phase function" (APF)

$P_R$ for "Rayleigh phase function" (RPF)

$\rho$ for "altitude normalized radiance" (ANR)

$y$ for "aerosol scattering index" (ASI)

5. (Time of day question) This statement was meant to reference dependence on scattering angle, so the reference now refers to "solar zenith angle" instead of time of day.

8. (Phase function figure question) We have added Fig. 6, which shows how the aerosol phase function varies during the year for several latitude bands.

9. (Fig. 11 reference question) This now refers to Fig. 16 - we have changed the reference so it refers to Sect. 3.4 rather than a particular future figure.

10. (Tangent altitude question) As now noted in the text, Fig. 11 (now Fig. 16) refers to $h = 25.5$ km and $h_n = 40.5$ km.

11. (Figs. 13-14 question) We have rearranged the text and these figures (formerly Figs. 13-15, now Figs. 18-20) to hopefully make the presentation clearer.

[Figure]

12. (Averaging kernel question) We have now (hopefully) addressed this point in the text.

13. (Fig. 19 question) This figure has been moved to its more logical place in the text (Sect. 3.3.2, now Fig. 15), and discussed at that point.
* * *
[Figure]

[revised manuscript text omitted]

 sensitivity to perturbations of the , using the V1  as a baseline. Note that the

[Figure]

**Figure 25.** $LER$ retrieved from radiances at $h = 40$ km (blue line) and 50 km (green line). Center slit observations from orbit 20234 are used in this example. Again, as noted in Fig. 16, the OMPS LP  aerosol extinction retrieval is insensitive to $LER$.

[Figure]

**Figure 26.** OMPS LP V1 (blue line) and OSIRIS V5.07 (red line) retrieved  aerosol extinction daily zonal means at selected altitudes from 2012 to 2016, at latitudes between $10°$ S and $0°$ S. The OSIRIS data set reports  aerosol extinction at 750 nm, so the OMPS  aerosol extinction was converted from 674 nm to 750 nm by using the angstrom coefficient consistent with the  aerosol size distribution assumed in the OMPS LP V1 algorithm.

[Figure]

**Figure 27.** Monthly zonal mean  aerosol extinction profiles at 750 nm derived from CALIPSO, OSIRIS and OMPS LP measurements during the aftermath of the Kelut eruption in 2014.

**Table 1.** Aerosol optical properties and aerosol size distribution ($ASD$) assumed in the V1 OMPS LP  aerosol extinction retrieval.

| | |
|---|---:|
| Real aerosol refractive index | 1.448 |
| Imaginary aerosol refractive index | 0 |
| Aerosol  median radius (fine mode), $r_1$ | 0.09 $\mu$ m |
| Aerosol mode width (fine mode), $\sigma_1$ | 1.4 |
| Aerosol  median radius (coarse mode), $r_2$ | 0.32 $\mu$ m |
| Aerosol mode width (coarse mode), $\sigma_2$ | 1.6 |
| Aerosol coarse mode fraction, $f_c$ | 0.003 |
| Aerosol scattering cross-section (at 675 nm) | $1.50 \times 10^{-10}$ cm$^2$ |

**Table 2.**  Aerosol size distributions assumed in several recent LS  aerosol extinction retrieval algorithms.

| Mission | Source | $r_0(\mu m)$ | $\sigma$ | $\alpha(525/1020)$ |
|---|---|---:|---:|---:|
| OMPS (V0.5) | (Loughman et al., 2015) | 0.06 | 1.73 | 2.34 |
| OSIRIS (V5) | (Bourassa et al., 2007) | 0.08 | 1.6 | 2.44 |
| SCIAMACHY (V1.1) | (Von Savigny et al., 2015) | 0.11 | 1.37 | 2.82 |
| Nyaku | (Nyaku, 2016), fine mode | 0.05105 | 1.43833 | 2.07 |
| | (Nyaku, 2016), coarse mode | 0.2025 | 1.15 | |

---

## Author Response (AR2)

**Final corrections for **"The Ozone Mapping and Profiler Suite (OMPS) Limb Profiler (LP) Version 1 Aerosol Extinction Retrieval Algorithm: Theoretical Basis" by Robert Loughman et al.**

We appreciate the recommendations made by the reviewer and editor.    In most cases, these were typographical corrections, etc. that we have accepted.    We list below a few other cases that require a more detailed response:

Comments from Anonymous Reviewer #1:

We have removed the equation that refers to the gain matrix from the manuscript (rather than adding a more detailed discussion of this point).

Associate Editor's Comments:

- We have changed the SO (solar occultation) acronym to SOT (solar occultation transmission).

- We have updated the reference to SAGE accuracy and precision to clarify the point made by Thomason et al. (2010).

- We have corrected the number of occultations per day to 30, as recommended.

- Regarding the terminology associated with equation (1):    I believe that the current usage is correct, and consistent with the text in Sect. 2.2.2 of the Grainger reference:    That is, $r_i$ corresponds to the median radius for the function $dN/dr$ (which we give as equation 1), and the mode radius is given by $r_i * \exp(-\sigma^2)$.    This is also consistent with the comment submitted by E. Malinina on Oct. 10, 2017, which motivated us to correct the initial version of the paper's terminology that called $r_i$ the "mode radius" for this function.    She refers to other references to support her argument (although I did not check those references for consistency with Grainger's presentation).

If I'm reading the Grainger reference correctly, the mean, median and mode of the function $dN/d \ln r$ are all identical (unlike the function $dN/dr$, for which mode < median < mean).    I suspect that this difference accounts for the muddled terminology associated with "log-normal distributions" in general, since various authors choose different functions to express    a "log-normal distribution."    The situation reminds me of the "Planck function", which has a different shape, mode, etc. depending on whether you express it in terms of "energy per unit of wavelength interval" or "per unit of frequency interval."

- We have included reference to the variation of Angstrom coefficient with altitude.

- As noted above, we have removed the reference to the gain matrix for this algorithm.

- We have added definitions for $I_m$ and $I_c$

- We have removed the former Figs. 1-2 (instead simply referred to them in their original context).

- We have updated the caption for the former Fig. 16 (now Fig. 14), and corrected the axis labels for the former Fig. 19 (now Fig. 17).

- We noted another inconsistency in the current Figs. 20-22 (replaced the former x-axis label "Mode radius" with "Median radius", consistent with the earlier discussion).

[revised manuscript text omitted]

**1.1 Occultation measurements**

The primary global record of stratospheric aerosol abundance has been derived from solar occultation (SO)  measurements. (This kind of data will be indicated as "solar occultation transmission (SOT)," to avoid confusion with the notation for sulfur oxide gases.) The Stratospheric Aerosol Measurement (SAM) / Stratospheric Aerosol and Gas Experiment (SAGE) series of missions pioneered this technique, with the long-lived SAGE II instrument (1984-2005) providing a particularly valuable continuous data record (Russell and McCormick, 1989; McCormick and Veiga, 1992; Thomason et al., 1997).  An overview of the large variation of stratospheric aerosol optical depth during the SAM/SAGE time period can be found in Fig. 1 of Thomason et al. (2008). These SOT measurements provide unmatched altitude resolution, precision and accuracy for stratospheric aerosol monitoring: Transmission profiles are produced on a 0.5 km grid with estimated vertical resolution of 0.7 km (SAGE, 2002). The SAGE aerosol extinction coefficient $\beta_a$ retrieval has targeted accuracy and precision  $= 5\%$, and analysis of the Version 4 product indicates accuracy and precision performance on the order of 10% for the 15-25 km altitude range (
[revised manuscript text omitted]